# Prevalence and Evolutionary Characteristics of Bovine Coronavirus in China

**DOI:** 10.3390/vetsci11060230

**Published:** 2024-05-21

**Authors:** Siyuan Li, Jin Huang, Xuhang Cai, Li Mao, Lingling Xie, Fu Wang, Hua Zhou, Xuesong Yuan, Xinru Sun, Xincheng Fu, Baochao Fan, Xingang Xu, Jizong Li, Bin Li

**Affiliations:** 1Institute of Veterinary Medicine, Jiangsu Academy of Agricultural Sciences, Nanjing 210014, China; 2021055552@nwafu.edu.cn (S.L.); 2021807135@stu.njau.edu.cn (J.H.); 2021055567@nwsuaf.edu.cn (X.C.); 20100014@jaas.ac.cn (L.M.); 2022807173@stu.njau.edu.cn (X.Y.); sxinruru@sina.cn (X.S.); fanbaochao.0405@163.com (B.F.); 2Key Laboratory of Veterinary Biological Engineering and Technology Ministry of Agriculture, Jiangsu Key Laboratory for Food Quality and Safety-State Key Laboratory Cultivation Base of Ministry of Science and Technology, Nanjing 210014, China; 3College of Veterinary Medicine, Northwest A&F University, Xianyang 712100, China; 4College of Veterinary Medicine, Nanjing Agricultural University, Nanjing 210095, China; 5Institute of Life Sciences, School of Food and Biological Engineering, Jiangsu University, Zhenjiang 212013, China; 6Jiangsu Co-Innovation Center for the Prevention and Control of Important Animal Infectious Disease and Zoonose, Yangzhou University, Yangzhou 225009, China; 7Guizhou Testing Center for Livestock and Poultry Germplasm, Guiyang 550018, China; xielingling0723@163.com (L.X.); cdzxwf@163.com (F.W.); 8Qianxi Animal Disease Control Center, Qianxi 551500, China; cdzxxll@163.com; 9Langfang Municipal Bureau of Agriculture and Rural Affairs, Langfang 065000, China; lfdj2008@126.com

**Keywords:** bovine coronavirus, cattle, diarrhea, genotypes, genetic evolution

## Abstract

**Simple Summary:**

Bovine coronavirus (BCoV), bovine rotavirus, bovine viral diarrhea virus, and bovine astrovirus are the most common intestinal pathogenic viruses causing diarrhea in cattle. BCoV was the major pathogen detected, with a positive rate of 34.02% (560/1646) from January 2020 to August 2023. The asymptomatic BCoV-infected rate (351/1646) was higher than the diarrhea BCoV-infected rate (209/1646). Studying the relevance of diarrhea associated with BCoV has shown that the onset of diarrheal symptoms post-infection is strongly correlated with the cattle’s age and may also be related to the breed. One the other hand, we amplified and sequenced the hemagglutinin esterase (HE), spike protein, and whole genomes of the partially positive samples and obtained six complete HE sequences, seven complete spike sequences, and six whole genomes. The results of molecular characterization revealed that six strains were in the GⅡb subgroup, and HBSJZ2202 and JSYZ2209 had four amino acid insertions on HE. We also analyzed ORF1a and found disparities across various regions within GIIb, which were positioned on separate branches within the phylogenetic tree. This work provides data for further investigating the epidemiology of BCoV and for understanding and analyzing BCoV distribution and dynamics.

**Abstract:**

Bovine coronavirus (BCoV), bovine rotavirus, bovine viral diarrhea virus, and bovine astrovirus are the most common intestinal pathogenic viruses causing diarrhea in cattle. We collected 1646 bovine fecal samples from January 2020 to August 2023. BCoV was the major pathogen detected, with a positive rate of 34.02% (560/1646). Of the 670 diarrheal samples and 976 asymptomatic samples, 209 and 351 were BCoV-positive, respectively. Studying the relevance of diarrhea associated with BCoV has shown that the onset of diarrheal symptoms post-infection is strongly correlated with the cattle’s age and may also be related to the breed. We amplified and sequenced the hemagglutinin esterase (HE), spike protein, and whole genomes of the partially positive samples and obtained six complete HE sequences, seven complete spike sequences, and six whole genomes. Molecular characterization revealed that six strains were branched Chinese strains, Japanese strains, and partial American strains from the GⅡb subgroup. Strains HBSJZ2202 and JSYZ2209 had four amino acid insertions on HE. We also analyzed ORF1a and found disparities across various regions within GIIb, which were positioned on separate branches within the phylogenetic tree. This work provides data for further investigating the epidemiology of BCoV and for understanding and analyzing BCoV distribution and dynamics.

## 1. Introduction

Coronavirus (CoV) belongs to the Coronaviridae family, which includes four genera: Alphacoronavirus, Betacoronavirus, Gammacoronavirus, and Deltacoronavirus [1]. Betacoronavirus is important to humans because it includes severe acute respiratory syndrome-related coronavirus, Middle East respiratory syndrome-related coronavirus, and severe acute respiratory syndrome coronavirus 2 (SARS-CoV-2) [2]. Human Betacoronavirus HCoV-OC43 was first isolated from a patient with respiratory disease in the 1960s [3], and BCoV was first described by Mebus et al. in the early 1970s as a cause of neonatal calf diarrhea [4]. Bovine coronavirus (BCoV) is involved in the etiology of at least three distinct clinical syndromes: enteric disease with high mortality in neonatal calves, winter dysentery with hemorrhagic diarrhea in cattle, and respiratory tract infections in cattle of all ages [5]. In 2007, Park et al. provided the first detailed report of dual enteric and respiratory tropisms of WD-BCoV in calves. Upon histopathological examination, BCoV-inoculated calves exhibited mild to severe villous atrophy, and the epithelia of the alveoli, bronchi, and bronchioles often appeared desquamated or necrotic [6]. Statistical analysis of the presence of BCoV in the feces of healthy calves and those with diarrhea showed that the BCoV detection rate in healthy calves ranged from 0 to 46% and that diarrhea in calves ranged from 3.4 to 69.0% [7,8,9]. The BCoV detection rate tended to be higher in calves with diarrhea than in healthy calves.

The virulence of BCoV related to calf diarrhea was clinically determined in Qinghai- Tibet Plateau, China, in 2019 [10]. Many studies have revealed that BCoV may cause or exacerbate calf diarrhea by co-infection with other enteric viruses such as bovine rotavirus (BRV) [11], bovine astrovirus (BoAstV), bovine torovirus (BToV), bovine viral diarrhea virus (BVDV), BCoV, and bovine kobuvirus (BKoV) (Cho et al., 2013). However, the prevalence of BCoV in diarrheal cases varies in these reports, at 69.05% in China (He et al., 2019; Keha et al., 2019), 68.6% in Brazil [11], 57.7% in Japan [12], 39.8% in South Korea [13], 21.9% in Turkey [14], and 7.2% in Iran [15]. BCoV can infect and spread to many animals, including sheep, musk, oxen, elk, sambar, deer, goats, dromedaries, camels, alpacas, giraffes, and wisents [2,16].

BCoV serum antibody test results have revealed differences between countries. The positive rates have ranged from 19 to 86% in Sweden [17,18,19,20], 48% to 53% in the United States [21], 30.0% in Belgium [22], 72.6% in Poland [23], 97.89% in Thailand [24], 12.82% in Inner Mongolia, China [25], and 30.8% in Campania, Italy [26]. In summary, BCoV is widespread in all regions, and persistent low-dose detoxification occurs, even when herds show no clinical symptoms [27].

At the molecular level, CoVs are positive-sense, single-stranded, enveloped RNA viruses [28]. Their genomes start from a 5′ untranslated region (5′ UTR), followed by two open reading frames (ORFs) 1a and 1b, four or five structural proteins, i.e., spike protein (S), membrane protein (M), nucleocapsid protein (N), hemagglutinin-esterase protein (HE), and envelope protein (E), and a 3′ untranslated region (3′ UTR). According to the CoV classification criteria proposed by the International Committee on the Taxonomy of Viruses in 2012, the four genera are further classified into different genotypes based on the mean amino acid genetic distance.

Although BCoV is one of the earliest discovered CoVs, its genome remained uncharacterized until it was sequenced in the US in 2001 [29]. To date, 98 genome sequences have been published in GenBank. However, few studies have addressed the prevalence and genetic diversity of BCoV and its association with ruminant diarrhea, likely owing to the difficulties in isolating this virus from cell and tissue cultures and the high proportion of co-infections in both diarrheal and asymptomatic animals. Fortunately, next-generation sequencing (NGS) [30] has effectively replaced conventional cell cultures to identify BCoV and other co-infected viruses. Thus, we conducted this study to determine the correlation between BCoV and bovine diarrhea by using NGS to investigate the prevalence and genomic characteristics of BCoV in Chinese cattle with and without diarrhea using recent samples collected from various regions. We conducted molecular characterization of HE, S, and the whole genome in the identified BCoV strains. This genetic evolution analysis revealed the evolutionary direction of BCoV over the past 3 years and provides a reference for subsequent research.

## 2. Materials and Methods

### 2.1. Sample Collection and Processing

We randomly collected 1646 fecal samples from five cattle breeds in 14 provinces and 30 provincial regions of mainland China from 2020 to 2023. Of these, 599 samples were sent by farmers whose cattle had diarrhea; the others were randomly collected from feces of asymptomatic cattle. All samples were freshly collected and then immediately transported to the laboratory and stored at −80 °C until use. All 1646 samples were tested for BCoV via RT-PCR. During laboratory analysis, frozen samples were thawed at room temperature, and samples of the same weight were transferred into 2 mL tubes and thoroughly vortexed with 500 μL of sterile phosphate-buffered saline. The resultant samples were subjected to two freeze–thaw cycles at −80 °C and room temperature to release the viral particles and then centrifuged at 12,000 rpm/min at 4 °C for 5 min. The supernatant was collected and filtered through 0.22 μm column filters to remove bacteria and contaminants.

### 2.2. Viral Detection

RNA was extracted using the FastPure Cell/Tissue Total RNA Isolation Kit V2 (Vazyme Biotech Co., Ltd, Nanjing, China) per the manufacturer’s instructions. The RNA was then mixed with HisScripII Q RT SuperMix for qPCR (Vazyme Biotech Co., Ltd, Nanjing, China) at a 1:4 ratio. RT-PCR was performed to amplify BCoV, BoRV, BVDV, and BoAstV [31] using the Green Taq Mix (Vazyme Biotech Co., Ltd, Nanjing, China). The PCR reaction volume was 25 μL (12.5 μL Green Taq Mix, 10 μmol/μL forward primer, 10 μmol/μL reverse primer, 2 μL of extracted RNA, and 8.5 μL of RNase-free water). For the negative controls, 2 μL of RNase-free water without RNA was added. The cycling parameters were 35 cycles of 95 °C for 15 s, 51 °C for 15 s, and 72 °C for 30 s, followed by a final extension at 72 °C for 5 min. The amplification products were detected by electrophoresis in 1.2% agarose gels.

### 2.3. BCoV Genome Sequencing

The S and HE genes were further sequenced using primers designed by Zhu et al. [32] and Park et al. [33], respectively. Other fragments using primers were based on He et al. [10]. The fragments with weak bands were cloned into the pMD19-T vector, and all segments were sequenced by Sangon Biotech Co., Ltd. (Shanghai, China). The nearly complete genome sequences of six BCoV strains were obtained by assembling the fragment sequences acquired above. To verify the results of the sequence assembly via NGS, all sequenced nucleotide positions were spliced by Seqman (DNAstar 7.0, DNASTAR, Inc, Madison, WI, USA, 2006) and confirmed by two or more independent sequencing reactions in both directions.

### 2.4. Genomic Analysis, Sequence Comparison, and Phylogenetic Analysis

The positions of the S and HE proteins were determined by NCBI (https://www.ncbi.nlm.nim.nih.gov/protein/, accessed on 6 April 2024) for amplification and use in this study. The S and HE proteins were predicted using SWISS-MODEL (https://swissmodel.expasy.org/, accessed on 6 April 2024) based on previously published sequences. The tertiary structure obtained from SWISS-MODLE was visualized using PyMOL 2.6.0a0 Open-Source (https://www.pymol.org/, accessed on 6 April 2024).

Multiple sequences were aligned using the Clustal W method with MEGA11.0.13. Maximum likelihood analysis was performed on the nucleotide and six amino acid sequences in different genomic regions of the BCoV strains. GeneDoc was used to optimize the alignment sequences. For the phylogenetic analysis, the ORF1a, S, HE, 4.8-kDa non-structural protein, 4.9-kDa non-structural protein, E, M, N, and the whole-genome sequences were aligned with reference sequences retrieved from GenBank. The unrooted phylogenetic trees were constructed using MEGA11.0.13 with bootstrap values calculated for 1000 replicates. iTOL optimization was performed based on the MEGA11.0.13 results.

### 2.5. Statistical Analysis

Statistical significance was determined by multiple-comparison *t*-tests and performed using univariate analysis by calculating the chi-square and analysis of variance with Tukey’s multiple-comparison tests using GraphPad Prism 5 software (GraphPad Software, Inc., San Diego, CA, USA). Direct comparisons between groups were made using lines. Asterisks denote statistical significance, with * *p* < 0.05, ** *p* < 0.01, and *** *p* < 0.001.

## 3. Results

### 3.1. BCoV Prevalence in Diarrheal and Asymptomatic Cattle

Samples were collected from 30 regions in 14 provinces of China throughout the year, covering all months. The 1646 samples primarily included cattle from Hebei (*n* = 406), Jiangsu (*n* = 459), Guizhou (*n* = 230), and Xizang (*n* = 271). Diarrhea prevalence was higher during the cold season than during the hot season (Appendix A).

From the 1646 samples, 40.70% (670/1646) of the cattle exhibited diarrhea symptoms, and 59.30% (976/1646) were asymptomatic. Among the BCoV-positive cattle (*n* = 560), 62.68% (351/560) were asymptomatic, and only 37.32% (209/560) of the cattle with diarrhea were BCoV-positive. This suggests that BCoV can exist in both healthy and diseased cattle. The overall prevalence of BCoV in all samples was 34.02% (560/1646), indicating an association between BCoV and cattle diarrhea (*p* = 0.045, odds ratio: 0.807, 95% confidence interval: 0.655–0.995; Table 1).

Age and breed information were collected to further analyze the factors contributing to BCoV infection in cattle. Most samples were from cattle older than 12 months (40.10%, 660/1646), and the highest proportion of diarrheal cases occurred in cattle younger than 3 months and accounted for 50% (113/205) of the reported cases. BCoV was significantly associated with diarrhea in this younger age group but was not correlated with cattle aged over 12 months (*p* = 0.483). The incidence of diarrhea after BCoV infection was higher in cattle younger than 3 months (Table 2).

BCoV infection rates varied among breeds (Table 3). Aberdeen Angus had the highest infection rate (72.22%, 39/54), followed by Yak (48.44%, 155/320), Simmental (32.00%, 80/150), domestic cattle (30.00%, 33/110), and Holstein (27.08%, 247/912). This suggests that Holstein cattle have a higher likelihood of developing diarrhea due to BCoV infection. However, this could potentially be attributed to the correlation inherent in sampling.

Notably, BCoV was detected in asymptomatic cattle across different age groups and breeds. Asymptomatic cattle aged 6–12 months had the highest detection rate within a 4-month period, accounting for 46.46% (177/381) of the samples. Among the five breeds, Yak had the highest BCoV detection rate among asymptomatic cattle, accounting for 49.36% (155/314) of the cases.

### 3.2. Co-Infection Analysis

Of the 1646 samples examined, 132 (8.02%) exhibited co-infections. BCoV was the primary pathogen in these cases. Notably, 12 cases exhibited triple infections (2 with BCoV, BRV, and BoAstV and 10 with BCoV, BVDV, and BoAstV). Eight distinct co-infection patterns were detected across all samples.

Of the 132 co-infection samples, 73 were associated with diarrhea, while the remaining samples appeared to be asymptomatic. BCoV was implicated in 117 cases of co-infection, but BCoV co-infection was unassociated with cattle diarrhea. Notably, analysis of the cattle with diarrhea revealed five co-infection events. The most prevalent combination was BCoV + BRV, accounting for 46.58% of the cases, followed by BCoV + BVDV (20.55%), BCoV + BoAstV (20.55%), BCoV + BRV + BoAstV (2.74%), and BCoV + BVDV + BoAstV (6.85%). The BCoV + BRV + BoAstV combination was not detected among the asymptomatic samples (Table 4).

From the co-infection samples, the most common co-infection was BCoV + BoAstV, accounting for 38.64% of all co-infections, suggesting the complex nature of co-infections involving BCoV. Co-infection was unassociated with cattle health but was associated with BRV (*p* = 0.000) and BoAstV (*p* = 0.000), particularly in cases of diarrhea (Table 1).

### 3.3. General Verification and Evolution Analysis

Nearly complete genomes were isolated from six strains from five areas (including seven diarrhea cases and two asymptomatic cases) via NGS and SeqMan. To verify the genome sequences obtained via NGS from the sample assembly, we performed additional RT-PCR to verify and assemble the whole genome, excluding the 3′ and 5′ terminal sequences. The results of the re-assembled genome sequences were completely consistent with those obtained via NGS.

Maximum likelihood analysis of the phylogenetic tree construction was based on the S gene and whole genome. The strains were isolated without changing the genotype, and all strains belonged to the GIIb subgroup with most American and Asian strains. The S genes of strains HBSJZ2202 and JSYZ2209 were much closer to strains SWUN/A1/2018 and SWUN/A10/2019 than to other strains. Phylogenetic tree analysis via the maximum likelihood method based on the HE gene indicated that the phylogenetic tree was divided into two main branches, and the insertion strains and deletion strain cluster were on a small branch. The evolutionary relationships of strains HBSJZ2202, JSYZ2209, and SWUN/A10/2018 were closer than were the other strains because they were clustered with GIIb genotypes and contained four amino acid insertions in the same site of the lectin domain. However, the S phylogenetic tree revealed that HBSJZ2202 and JSYZ2209 were also clustered with SWUN/A10/2018, as was HE (Figure 1).

We aligned ORF1a, 4.8k-Da non-structural protein, 4.9k-Da non-structural protein, E, M, and N. We noted amino acid polymorphism within the genes encoding ORF1 and four amino acid deletions on strains ON142317 and ON093194 sequenced from China in 2021. Notably, except for the Yak strain, the remaining Chinese strains were independent of other GIIb subtypes on an independent branch. The known Chinese strains have three mutation sites at amino acid sites 834, 857, and 889 (Figure 2). Other genotypes also exhibit distinctions; however, whether the variations at these amino acid locations are significant remains uncertain. Additionally, the Chinese strains contain many unique sites; however, whether these sites are distributed in different non-structural protein regions of BCoV is unclear.

The HE homologies of these six strains ranged from 97 to 99.99% (Table 5), and the S homologies of seven strains ranged from 98.5 to 100% (Table 6). JSYZ2209 and HBSJZ2202 had the highest homology for both the HE and S genes. For the whole genomes of the six strains, the homology ranged from 98.8 to 99.6% (Table 7).

### 3.4. Putative Protein Analysis

We used SWISS-MODEL to establish the 3D model of HE based on the BCoV crystal model (PDB:3c14) and PyMOL to analyze the structure. Four amino acid insertions altered the crystal structures of strains JSYZ2209 and HBSJZ2202. Specifically, the four amino acid insertions at positions 212–215 changed from beta folding/random crimping/beta folding to random crimping/alpha helix/random crimping (Figure 3). Positions 211F, 212L, 213S, and 214N play critical roles in ligand–protein interactions in BCoV HE [34], but an amino acid insertion at 212KATV216 and deletion at 208NGFK211 have been reported, indicating that the BCoV-HE ligand can change.

## 4. Discussion

Juvenile animals are usually more susceptible to diarrhea-causing viruses and present obvious symptoms, and our results were consistent with this. However, under the conditions of non-intensive management, infected older animals tend to have more fatalities even if they exhibit no signs of diarrhea. We found 351 cases of asymptomatic infections, primarily occurring in cattle older than 6 months. Conversely, most samples with diarrheal infections were from younger cattle, primarily under 6 months of age. Cattle without diarrhea symptoms posed a higher risk, particularly if they were older. Most of our samples were collected from non-intensively managed farms. Inadequate disinfection and protective measures allow diseases to spread much easier. The relatively poor environment and substandard prevention and control measures may have led to the BCoV epidemic among farms.

The high prevalence of BCoV may be similar to that of the porcine epidemic diarrhea virus, which caused a regional epidemic [33,35]. Previous reports stated that BCoV that caused diarrhea in cattle was often accompanied by co-infection with other pathogens [11,12,13]. In this study, the infection rate of cattle with diarrhea and co-infected with four other viruses reached 7.96% (excluding other diarrhea-causing pathogens). Therefore, the co-infection rate was actually much higher. Whether BCoV infection causes diarrhea in cattle remains uncertain. Additionally, BCoV has great potential for cross-species transmission. Previous studies report that BCoV infects the respiratory tract, followed by the digestive tract, thus causing diarrhea [35,36]. Similar to SARS [37,38] and SARS-2 [39], BCoV has two tissue tropisms. Szczepanski et al. indicated that BCoV can attach to HLA-I from humans, and HLA-I may be a candidate protein receptor [40], the entry receptor of OC43 on some cells [41,42,43], or the accessory receptor of HKU1 [44]. Therefore, BCoV is a good model for exploring other betacoronaviruses.

This study yielded a high positive rate for BCoV. Importantly, infected asymptomatic cattle accounted for 51.39%, indicating that the infection rate of BCoV increased from 2020 to 2022 and is widely distributed in cattle. Infected cattle can discharge BCoV into the surrounding environment, and the virus can persist in low temperatures and high relative humidity for 120 h and in feces for ≥96 h [45]. During this study, cattle were infected, and some exhibited diarrhea, while others did not. However, calves were more susceptible to developing diarrhea as a result of the infection. Furthermore, freezing temperatures can aggravate diarrhea, easily causing winter dysentery. However, other studies have shown that BCoV-induced diarrhea was not correlated with low temperatures [30,33,46].

HE and S are critical for BCoV invasion and cellular release and are critical antibody-neutralization epitopes. Zeng et al. [34] showed that BCoV HE was similar to the HA of IV but was more flexible than HA and was limited to S. S and HE can attach to the same sialid acid, 9-O-acetylated sialoglycan [47,48,49]. Lang et al. indicated that HE and S are whole, and when HE incurs a critical mutation, it induces S mutations and strengthens its affinity [50]. Interestingly, HE forms large branches in the phylogenetic tree and insertions from American strains independent of those from China and different G-type strains. HE sequences are distributed between the two branches, but S is clearly separated. On S, S1 is a highly mutated region, and S2 is relatively conserved. Previous reports revealed co-evolution between the paired amino acids [35,51], and the appearance of GIIa subgroups may be associated with co-evolution between amino acids of the same protein. S structural changes are induced by host-receptor binding. The key to this transformation depends on their stability, and S1 shedding is considered a prerequisite [52]. According to Guan et al., BCoV may use the two-receptor binding motif, in which S1-NTD attaches to 9-O-acetylated sialoglycans, and S1-RBD may function as the protein receptor [53]. Thus, mutations at receptor-binding positions on S1 are particularly critical and will impact the affinity and modify and alter the types of binding receptors.

Insertions and deletions in HE sequences have been reported from both China and the United States [5,10,54,55]. Samples isolated from Liaoning, China, revealed four amino acid insertions (GenBank accession number MK095142). We previously detected no HE insertions until 2022, when HBSJZ2202 and JSYZ2209 were collected from Hebei and Jiangsu, respectively. A report from 4 years ago revealed that HE insertions were detected in various regions, which may have attributed to the recombination of strains. These mutations occurred in a critical position on the glycan receptor and changed its structure [34]. Workman et al. suggested that insertions and deletions do not abolish binding receptors [5]. Insertions and deletions are the most important mutations of BCoV HE. Notably, the ability of binding receptor 9-O-acetylated sialoglycans is stronger than that of insertions and deletions. However, Langereis et al. showed that moderately changing the structure of MHV HE changed the glycan tropism. Therefore, BCoV-HE mutations may be similar to those of MHV [56]. The sequences with HE mutations on the phylogenetic trees were on a small branch, and S was on a closer branch. This is consistent with the idea of co-evolution. Whether the relationship between HE mutations and S is associated with BCoV prevalence is unknown; however, BCoV is widely distributed in parts of China.

Alignment and the phylogenetic tree revealed that the genetic diversity of BCoV was reflected in HE and S, as well as in other structural and non-structural proteins. A previous report found that truncation of a large-scale genome encoding 4.8-kDa non-structural and 4.9-kDa non-structural proteins was associated with adaptation to human hosts [57]. Although this truncation differs from that of CoV-OC43, truncation of the 4.8-kDa non-structural and 4.9-kDa non-structural proteins may have been associated with adaptation to bovines or other hosts.

Our research revealed the prevalence and detailed molecular characterization of BCoV and indicated the increasing infection rate of asymptomatic cattle. The occurrence of diarrhea following BCoV infection is significantly correlated with herd age and may be influenced by breed. HE mutations detected from the same area can emerge from nothing and may reflect this pattern in the evolution of BCoV. We analyzed the ORF1a gene. Despite belonging to the same genotype, the Chinese strain was identified in a separate branch along with other GIIb genotypes on ORF1a, suggesting differences in the non-structural proteins. Our results offer insight into diarrhea-associated BCoV, which demonstrates higher susceptibility among calves. Notably, the Chinese strain of the GIIb genotype exhibited distinct differences within the ORF1a region compared with that of other regions, providing important information for future research.

## Figures and Tables

**Figure 1 vetsci-11-00230-f001:**
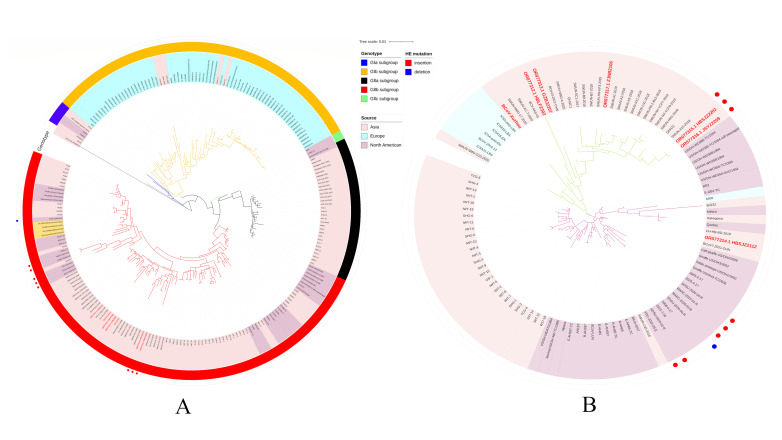
Phylogenetic analysis by maximum likelihood of BCoV strains. Red bold font indicates sequences isolated in this study. (**A**) Phylogenetic analysis of the global BCoV strains based on S gene. (**B**) Phylogenetic analysis of the global BCoV strains based on HE gene. Red label: the sequence was obtained in this laboratory.

**Figure 2 vetsci-11-00230-f002:**
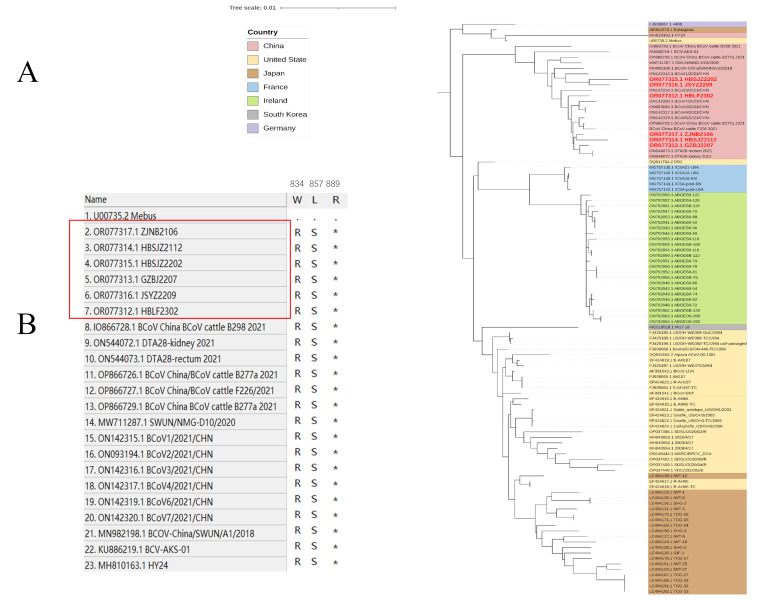
Phylogenetic analysis by maximum likelihood of BCoV strains. Red bold font indicates sequences isolated in this study. (**A**) Phylogenetic analysis of the global BCoV strains based on the ORF1a gene. (**B**) Unique amino acid sequence site of Chinese strain. Red label: the sequence was obtained in this laboratory (*: termination codon).

**Figure 3 vetsci-11-00230-f003:**
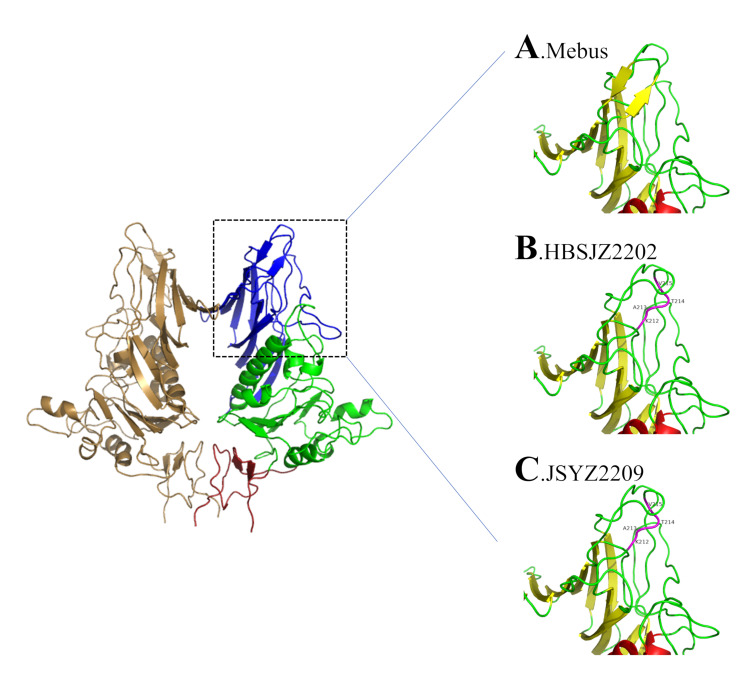
HE putative structure. (**A**) Mebus putative structure of HE in lectin domain. (**B**) BCoV-SJZ2 putative structure of HE in lectin domain. (**C**) BCoV-YZ1 putative structure of HE in lectin domain. The domains are color-coded: lectin domain (R, blue); esterase domain (E, green); membrane-proximal domain (MP, red); helix (red); sheet (yellow); loop (green); insertion position (purple).

**Table 1 vetsci-11-00230-t001:** Correlation between co-infection and diarrhea.

	Diarrheal	Asymptomatic	Total	OR (95%CI: Lower, Upper)	*p*-Value
BCoV positive	209	351	560	0.807 (0.655~0.995)	0.045
BCoV negative	461	625	1086		
Total	670	976	1646		
BCoV co-infection	64	53	117	0.805 (0.269~2.407)	0.698
BCoV negative	9	6	15		
Total	73	59	132		
BRV positive	210	30	240	14.396 (9.665~21.442)	0.000
	**Diarrheal**	**Asymptomatic**	**Total**	**OR (95%CI: lower, upper)**	***p*-value**
BRV negative	460	946	1406		
Total	670	976	1646		
BRV co-infection	42	12	54	5.306 (2.419~11.641)	0.000
BRV negative	31	47	78		
Total	73	59	132		
BoAstV positive	50	59	109	1.253 (0.848~1.852)	0.257
BoAstV negative	620	917	1537		
Total	670	976	1646		
BoAstV co-infection	31	45	76	0.230 (0.108~0.490)	0.000
BoAstV negative	42	14	56		
Total	73	59	132		
BVDV co-infection	17	13	30	1.074 (0.474~2.441)	0.864
BVDV negative	56	46	102		
Total	73	59	132		

**Table 2 vetsci-11-00230-t002:** Infection at different months old.

Months Old		Diarrhea	Positive Number	Asymptomatic	Positive Number	*p*-Value	OR (95%CI: Lower, Upper)
0–3	Holstein	170	100	76	16	0	4.606 (1.665~4.117)
	Domestic cattle	20	4	0	0		
	Simmental	9	9	0	0		
	Aberdeen Angus	0	0	0	0		
	Yak	6	0	0	0		
Total		205	113	76	16		
3–6	Holstein	126	44	31	18	0	0.19 (0.093~0.389)
	Domestic cattle	0	0	0	0		
	Simmental	46	12	0	0		
	Aberdeen Angus	0	0	15	15		
	Yak	0	0	0	0		
**Months old**		**Diarrhea**	**Positive number**	**Asymptomatic**	**Positive number**	***p*-value**	**OR (95%CI: lower, upper)**
Total		172	56	46	33		
6–12	Holstein	100	17	34	8	0	0.319 (0.193~0.529)
	Domestic cattle	0	0	25	12		
	Simmental	6	6	126	39		
	Aberdeen Angus	0	0	39	24		
	Yak	0	0	157	94		
Total		106	23	381	177		
>12	Holstein	187	17	188	27	0.483	0.851 (0.542~1.336)
	Domestic cattle	0	0	65	17		
	Simmental	0	0	63	20		
	Aberdeen Angus	0	0	0	0		
	Yak	0	0	157	61		
Total		187	17	473	125		

**Table 3 vetsci-11-00230-t003:** Infection at different breeds.

Breeds	Months Old	Diarrhea	Positive Number	Asymptomatic	Positive Number	*p*-Value	OR (95%CI: Lower, Upper)
Holstein	0–3	170	100	76	16	0	2.907 (2.080~4.062)
	3–6	126	44	31	18		
	6–12	100	17	34	8		
	>12	187	17	188	27		
Domestic cattle	0–3	20	4	0	0	0.286	0.526 (0.161~1.714)
	3–6	0	0	0	0		
	6–12	0	0	25	12		
	>12	0	0	65	17		
Simmental	0–3	9	9	0	0	0.641	1.157 (0.628~2.132)
	3–6	46	12	0	0		
	6–12	6	6	126	39		
	>12	0	0	63	20		
Aberdeen Angus	0–3	0	0	0	0	0.644	0.392 (0.007~20.661)
	3–6	0	0	15	15		
	6–12	0	0	39	24		
	>12	0	0	0	0		
Yak	0–3	6	0	0	0	0.084	0.079 (0.004~1.412)
	3–6	0	0	0	0		
	6–12	0	0	157	94		
	>12	6	0	0	0		

**Table 4 vetsci-11-00230-t004:** Percentage of 5 BCoV co-infection combination.

Diarrheal Calves			Asymptomatic Calves		
Patterns of Co-Infection	Sample No.	Percentages	Patterns of Co-Infection	Sample No.	Percentages %
BCoV + BRV	34	46.58%	BCoV + BRV	6	10.17%
BCoV + BVDV	8	10.96%	BCoV + BVDV	6	10.17%
BCoV + BoAst	15	20.55%	BCoV + BoAst	36	61.02%
BCoV + BRV + BoAst	2	2.74%	BCoV + BVDV + BoAst	5	8.47%
BCoV + BVDV + BoAst	5	6.85%	BoAst + BRV	4	6.78%
BoAstV + BVDV	3	4.11%	BVDV + BRV	2	3.39%
BoAstV + BRV + BVDV	1	1.37%			
BoAst + BRV	5	6.85%			
Total	73		Total	59	

**Table 5 vetsci-11-00230-t005:** Pairwise amino acid sequence identities (%) among six BCoV HE sequences in this study.

Strains	HBLF2302	HBSJZ2112	HBSJZ2202	GZBJ2207	ZJNB2106	JSYZ2209
Hemagglutinin Esterase						
HBLF2302	100.0	98.0	98.9	98.4	99.1	98.8
HBSJZ2112	2.0	100.0	97.1	98.0	97.5	97.0
HBSJZ2202	1.1	3.0	100.0	97.6	99.0	99.9
GZBJ2207	1.7	2.1	2.5	100.0	98.1	97.5
ZJNB2106	0.9	2.6	1.0	1.9	100.0	98.9
JSYZ2209	1.2	3.1	0.1	2.6	1.1	100.0

**Table 6 vetsci-11-00230-t006:** Pairwise amino acid sequence identities (%) among seven BCoV S sequences in this study.

Strains	HBLF2302	HBSJZ2112	HBSJZ2202	GZBJ2207	ZJNB2106	JSYZ2209	BCoV-JSXZ
Spike Protein							
HBLF2302	100.0	99.2	98.9	99.2	99.1	98.9	99.0
HBSJZ2112	0.8	100.0	98.9	99.6	99.2	98.9	98.8
HBSJZ2202	1.2	1.2	100.0	98.8	98.8	100.0	98.5
GZBJ2207	0.8	0.4	1.2	100.0	99.2	98.9	98.8
ZJNB2106	0.9	0.8	1.2	0.8	100.0	98.8	98.7
JSYZ2209	1.1	1.1	0.0	1.2	1.2	100.0	98.5
BCoV-JSXZ	1.0	1.2	1.6	1.2	1.3	1.5	100.0

**Table 7 vetsci-11-00230-t007:** Pairwise amino acid sequence identities (%) among six BCoV genome sequences in this study.

Strains	HBLF2302	HBSJZ2112	HBSJZ2202	GZBJ2207	ZJNB2106	JSYZ2209
Whole Genomes						
HBLF2302	100	99.3	98.9	99.3	99.6	99.1
HBSJZ2112	0.7	100	98.8	99.2	99.4	99.0
HBSJZ2202	1.1	1.2	100	98.9	99.0	99.4
GZBJ2207	0.7	0.8	1.1	100	99.4	98.8
ZJNB2106	0.4	0.6	1.0	0.6	100	99.2
JSYZ2209	0.9	1.0	0.6	1.2	0.8	100

## Data Availability

The raw data supporting the conclusions of this article will be made available by the authors upon request.

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
