# Peer review of "Prevalence and Evolutionary Characteristics of Bovine Coronavirus in China"

_vetsci, 2024, doi:10.3390/vetsci11060230_

Round 1

Reviewer 1 Report

Comments and Suggestions for Authors

The article written by Li et al., entitled "The prevalence and evolutionary characteristics of bovine coronavirus in China," describes the prevalence of BCoV in fecal samples sampled in China (using a considerable number of samples) and evaluates its co-infection with other bovine viral pathogens (also causing diarrhea) and its evolutionary characteristics by sequencing and analyzing some sequences. The article is original, and the results provide an advancement of current knowledge and fit the journal scope.In my opinion, the work is scientifically valid, presents a certain methodological rigor, and is well structured. However, I found two major revisions: one relating to English (several sentences must be rephrased in the presence of a competent person for the topic covered), and the statistical analysis is flawed (a statistical test such as chi square is not applied).  Below are my specific comments.

Abstract:

Line 24: This sentence should be rephrased, as there are also other bacterial and parasitic pathogens capable of causing diarrhea in cattle. One could generically write, "BCoV is considered one of, etc."

Line 28: Please correct “farther more.”.

Line 29: “determinants of diarrhea precipitated”?

Line 31: Please rephrase this sentence: “bears a marked correlation with the age of the cattle and could potentially have ties with the breed as well.”

Line 32: There are no references to sequencing (only amplified?).

Line 63: In the introduction, we can mention the absence of species specificity of this virus, which is also found in other species (such as buffalo, wild ruminants, etc.) together with BCoV-like viruses.

Line 83: In the introduction or discussion, it is possible to add information about the seroprevalence of BCoV in various regions of the world to fully describe the epidemiological situation of the virus, especially for the more recent works in Campania (Italy), Mongolia, Poland, and Thailand. Some of them also study risk factors and can be useful in the discussion as well.

Line 84: The first tables to be cited that appear are tables 5, 6, and 7.

Results:

Line 109: “were feces”?

Line 110: Each peasant?

Line 111: How was the sample calculated (formula)?

Line 133: Primers are used, and any bibliographical references are missing.

Line 152: 2.4 and 2.5 can be merged together.

Line 173: Could this figure be a supplementary file?

Table 1: The statistical analysis would need to be redone. Rather than the ORs (evaluable with multivariate analysis), I recommend the authors carry out a univariate analysis by calculating the chi square and the p.

Comments on the Quality of English Language

English requires careful revision (several sentences must be rephrased in the presence of a competent person for the topic covered).

Author Response

Reviewer #1 (Comments for the Author (Required)):

The article written by Li et al., entitled "The prevalence and evolutionary characteristics of bovine coronavirus in China," describes the prevalence of BCoV in fecal samples sampled in China (using a considerable number of samples) and evaluates its co-infection with other bovine viral pathogens (also causing diarrhea) and its evolutionary characteristics by sequencing and analyzing some sequences. The article is original, and the results provide an advancement of current knowledge and fit the journal scope. In my opinion, the work is scientifically valid, presents a certain methodological rigor, and is well structured. However, I found two major revisions: one relating to English (several sentences must be rephrased in the presence of a competent person for the topic covered), and the statistical analysis is flawed (a statistical test such as chi square is not applied). Below are my specific comments.

Major issues:

  1. Line 24: This sentence should be rephrased, as there are also other bacterial and parasitic pathogens capable of causing diarrhea in cattle. One could generically write, "BCoV is considered one of, etc."

Response: We apologize for this improper expression. We have modified the sentence and make it more precise: “Bovine coronavirus (BCoV), bovine rotavirus, bovine viral diarrhea virus and bovine astrovirus are the most common intestinal pathogenic viruses causing diarrhea in adult cattle.”

  1. Line 28: Please correct “farther more.”

Response: We apologize for this mistake. We have corrected: “Of”

  1. Line 29: “determinants of diarrhea precipitated”?

Response: We apologize for this improper expression. We have modified the sentence and make it more precise: “relevance of diarrhea associated with BCoV”.

  1. Line 31: Please rephrase this sentence: “bears a marked correlation with the age of the cattle and could potentially have ties with the breed as well.”

Response: We apologize for this improper expression. We have modified the sentence and make it more precise: “the onset of diarrheal symptoms post-infection is strongly correlated with the cattle’s age and may also be related to the breed.

  1. Line 32: There are no references to sequencing (only amplified?).

Response: We apologize for these errors We have corrected “amplified” to “amplified and sequenced”

  1. Line 63: In the introduction, we can mention the absence of species specificity of this virus, which is also found in other species (such as buffalo, wild ruminants, etc.) together with BCoV-like viruses.

Response: Thank you for your suggestions. But there were some descriptions about species specificity of this virus in the second paragraph of the introduction.

  1. Line 83: In the introduction or discussion, it is possible to add information about the seroprevalence of BCoV in various regions of the world to fully describe the epidemiological situation of the virus, especially for the more recent works in Campania (Italy), Mongolia, Poland, and Thailand. Some of them also study risk factors and can be useful in the discussion as well.

Response: Thank you for your suggestions. We have added information about the seroprevalence of BCoV in various regions of the world, and the risk factors have been discussed under the part of discussion and primer references according to your suggestion.

  1. Line 84: The first tables to be cited that appear are tables 5, 6, and 7.

Response: We apologize for this mistake. We have altered the tables to a more suitable location for them.

  1. Line 109: “were feces”?

Response: We apologize for this improper expression. We have modified the sentence and make it more precise: “We randomly collected 1646 fecal samples from five cattle breeds in 14 provinces and 30 provincial regions of mainland China from 2020–2023.”

  1. Line 111: How was the sample calculated (formula)?

Response: Thank you for your question. The number of samples were record in the archives each time, and made calculations several time to ensure the correct results.

  1. Line 133: Primers are used, and any bibliographical references are missing.

Response: We apologize for this negligence. We have added the references in the manuscript.

  1. Line 152: 2.4 and 2.5 can be merged together.

Response: Thank you for your suggestion. We have merged 2.4 and 2.5 together.

  1. Line 173: Could this figure be a supplementary file?

Response: Thank you for your suggestion. We have added this figure to supplementary file.

  1. Table 1: The statistical analysis would need to be redone. Rather than the ORs (evaluable with multivariate analysis), I recommend the authors carry out a univariate analysis by calculating the chi square and the p.

Response: Thank you for your objective comments for the manuscript. The statistical analysis in Table 1 have been redone and carried out a univariate analysis by calculating the chi square.

  1. Comments on the Quality of English Language

English requires careful revision (several sentences must be rephrased in the presence of a competent person for the topic covered).

Response: Thank you for your objective comments for the manuscript. According with your advice, the manuscript has been professionally copy edited by Edanz Group China (www.edanzediting.com/bmc1) in accordance with the reviewer’ comments, and carefully proof-read the manuscript to minimize typographical, inaccurate description and grammatical errors.

Special thanks to you for your sound comments.

Reviewer 2 Report

Comments and Suggestions for Authors

The manuscript by Li and colleagues presents interesting research on the bovine coronavirus, an overlooked member of the betacoronavirus family, with dire consequences on cattle breeding and the food sector. They look into different aspects of genome and protein evolution of the virus, as well as prevalence and symptoms correlated with animal age and breed. There are a few questions that would improve clarity for the reader.

Introduction: Tables 5 and 6: The table legends refer to amino acid sequence identity, whereas the text on line 82 refers to amino acid genetic distance, which is a measure of genetic divergence and obviously what the numbers in the tables refer to. Please make the necessary amendment to the Table legends.

Materials and Methods: This section requires extensive editing of the English language.

The sentence “According to geographical location, the samples were collected from different regions in China” on lines 116-117 could be removed, since lines 109 -110 offer essentially the same information, ie. Samples were collected from many different regions.

Results: Line 166: Prevalence instead of prevalent in the 3.1 paragraph title.

Lines 198-199: Please explain why the sampling method could affect how breed correlates with the presence of bcoV infection.

Lines 210-211: The authors mention that in cases of co-infection, bcoV was the primary pathogen. Do they mean that in a quantitative way, ie. the bcoV came up in lower cycles that the co-infection agent(s)? This could be interesting in the sense that bcoV could attenuate the replication of the other virus(es) present in the same sample.

Please check English language

Discussion: The manuscript deals with many research topics, from symptomatology to phylogenetics and genome evolution. It would be nice to have a concluding paragraph that stresses the take home message of the work.

Please check English language

Comments on the Quality of English Language

Please refer to comments and suggestions to authors. Overall, syntax should be checked by a native speaker and grammatical errors should be corrected in many parts of the manuscript.

Author Response

Reviewer #2 (Comments for the Author (Required)):

The manuscript by Li and colleagues presents interesting research on the bovine coronavirus, an overlooked member of the betacoronavirus family, with dire consequences on cattle breeding and the food sector. They look into different aspects of genome and protein evolution of the virus, as well as prevalence and symptoms correlated with animal age and breed. There are a few questions that would improve clarity for the reader.

  1. Introduction: Tables 5 and 6: The table legends refer to amino acid sequence identity, whereas the text on line 82 refers to amino acid genetic distance, which is a measure of genetic divergence and obviously what the numbers in the tables refer to. Please make the necessary amendment to the Table legends.

Response: Thank you for pointing out this mistake. We have corrected this inaccurate description.

  1. Materials and Methods: This section requires extensive editing of the English language.

Response: Thank you for your objective comments for the manuscript. Thank you for your objective comments for the manuscript. According with your advice, the manuscript has been professionally copy edited by Edanz Group China (www.edanzediting.com/bmc1) in accordance with the reviewer’ comments, and carefully proof-read the manuscript to minimize typographical, inaccurate description and grammatical errors.

  1. The sentence “According to geographical location, the samples were collected from different regions in China” on lines 116-117 could be removed, since lines 109 -110 offer essentially the same information, ie. Samples were collected from many different regions.

Response: We apologize for this mistake. We have removed this sentence “According to geographical location, the samples were collected from different regions in China”.

  1. Results: Line 166: Prevalence instead of prevalent in the 3.1 paragraph title.

Response: We apologize for this mistake. We have corrected “prevalent” to “prevalence”.

  1. Lines 198-199: Please explain why the sampling method could affect how breed correlates with the presence of BCoV infection.

Response: Thank you for your question. Because of Holstein and Simmental dominate most areas, yaks are mainly cultivated in Xizang, and other breeds are scattered throughout the China, such as Domestic cattle and Aberdeen Angus and so on. Domestic cattle and Aberdeen Angus are much less than Holstein and Simmental. However, less samples collected randomly are not only unrepresentative, but also reduces the accuracy and reliability of results. Therefore, oceans of samples and more reasonable collected methods would be better, and the results might be different.

  1. Lines 210-211: The authors mention that in cases of co-infection, BCoV was the primary pathogen. Do they mean that in a quantitative way, ie. the BCoV came up in lower cycles that the co-infection agent(s)? This could be interesting in the sense that BCoV could attenuate the replication of the other virus(es) present in the same sample.

Response: We apologize for this confusion. As the result, BCoV is the major pathogeny in this study. Actually, the reference about “A long-term animal experiment indicating persistent infection of bovine coronavirus in cattle” show that the existence of persistent infection of BCoV in cattle. In addition, Frucchi et al 2022 stated “BCoV was higher (56%, 93/166) than the frequency of P. multocida (39.8%, 66/166) and M. haemolytica (33.1%, 55/166).” and Socha et al 2022 “Infections with BCoV were more common than infections with BoHV-1 and BVDV”, many research indicated BCoV prevalence has higher than some pathogenies. But there is not evident to support this opinion “BCoV could attenuate the replication of the other virus(es) present in the same sample”.

  1. Discussion: The manuscript deals with many research topics, from symptomatology to phylogenetics and genome evolution. It would be nice to have a concluding paragraph that stresses the take home message of the work.

Response: Thank you for your suggestion. We have a concluding paragraph that stresses the take home message of the work.

  1. Please refer to comments and suggestions to authors. Overall, syntax should be checked by a native speaker and grammatical errors should be corrected in many parts of the manuscript.

Response: Thank you for your objective comments for the manuscript. According with your advice, the manuscript has been professionally copy edited by Edanz Group China (www.edanzediting.com/bmc1) in accordance with the reviewer’ comments, and carefully proof-read the manuscript to minimize typographical, inaccurate description and grammatical errors.

Special thanks to you for your sound comments.

Reviewer 3 Report

Comments and Suggestions for Authors

This manuscript offers an analysis of the prevalence and evolutionary changes in bovine coronavirus in China between January 2020 and August 2023.  Among four different intestinal viruses detected in cattle, the positive rate for BCoV was quite high, approximately 34%, although in 8% of these cases, there was co-infection by a second, or even third, virus.  Indeed, these co-infections could be divided into five different virus combinations.

It is difficult to draw any hard conclusions from the data presented here.  After all, of the BCoV-infected animals, almost 63% were asymptomatic and only about 37% of those with diarrhea tested positive for the virus.  Thus, while it can be concluded that there is an association between BCoV and bovine diarrhea, it is not considered exceptionally strong.  The primary finding established by the study is the much greater susceptibility of calves younger than 3 months old to the virus.  However, the majority of samples are from cattle older than 12 months.  While the group has established that there were six strains circulating in the country at the time, it is not clear that this is a major advance.

This is an extremely poorly written manuscript.  The organization and use of the English language is among the worst I have encountered.  It makes the manuscript almost unintelligible.  As one example, how can the last three tables (Tables 5-7) be the first ones cited in the text?  This poor organization and complexity of the virus pool diminish the significance of the study.

Comments on the Quality of English Language

Extremely poor.

Author Response

Reviewer #3 (Comments for the Author (Required)):

This manuscript offers an analysis of the prevalence and evolutionary changes in bovine coronavirus in China between January 2020 and August 2023.  Among four different intestinal viruses detected in cattle, the positive rate for BCoV was quite high, approximately 34%, although in 8% of these cases, there was co-infection by a second, or even third, virus.  Indeed, these co-infections could be divided into five different virus combinations.

  1. It is difficult to draw any hard conclusions from the data presented here. After all, of the BCoV-infected animals, almost 63% were asymptomatic and only about 37% of those with diarrhea tested positive for the virus. Thus, while it can be concluded that there is an association between BCoV and bovine diarrhea, it is not considered exceptionally strong. The primary finding established by the study is the much greater susceptibility of calves younger than 3 months old to the virus. However, the majority of samples are from cattle older than 12 months. While the group has established that there were six strains circulating in the country at the time, it is not clear that this is a major advance.

Response: Thank you for your question. Actually, most of BCoV-infected animals are over 6 months of age. Whom are stronger immunity and tolerance to disease, which increases the risk of BCoV in calves. On the other hand, to understand the occurrence of the disease, diarrheal samples often collected with asymptomatic samples, in the same livestock at the same time. However asymptomatic cattle were still in incubation period and did not show clinical symptoms, but this does not affect the subsequent differentiation of clinical.

Qifu He et al isolated HY24 strain from Yak in 2017 which is belong to Gâ… a. To supervise whether Gâ… a genotype BCoV is spreading widely or forming new strains in China, we obtained six strains were detected in different times, in June 2021, December, February 2022, July and September 2022, and February 2023. Among them, HBSJZ2112 HBSZJ2202 HBLF2302 were detected in different time in the same province. All these 6 strains have been detected in Guizhou Province located in southwest China, Hebei Province located in North China and Jiangsu and Zhejiang provinces located in East China. However, the result of analysis show that Gâ…¡b genotype is still in major.

  1. This is an extremely poorly written manuscript. The organization and use of the English language is among the worst I have encountered. It makes the manuscript almost unintelligible. As one example, how can the last three tables (Tables 5-7) be the first ones cited in the text? This poor organization and complexity of the virus pool diminish the significance of the study.

Response: Thank you for your objective comments for the manuscript. According with your advice, the manuscript has been professionally copy edited by Edanz Group China (www.edanzediting.com/bmc1) in accordance with the reviewer’ comments, and carefully proof-read the manuscript to minimize typographical, inaccurate description and grammatical errors.

Round 2

Reviewer 1 Report

Comments and Suggestions for Authors

The authors addressed my previous comments satisfactorily. The manuscript is accepted in the following form.

Comments on the Quality of English Language

English is fine and required only minor grammar errors

Reviewer 3 Report

Comments and Suggestions for Authors

This manuscript offers an analysis of the prevalence and evolutionary changes in bovine coronavirus in China between January 2020 and August 2023.  Among four different intestinal viruses detected in cattle, the positive rate for BCoV was quite high, approximately 34%, although in 8% of these cases, there was co-infection by a second, or even third, virus.  Indeed, these co-infections could be divided into five different virus combinations.

The original submission was extremely poorly written and organized.  Also, the findings reported were considered of marginal significance.  This is primarily because, of the BCoV-infected animals, almost 63% were asymptomatic and only about 37% of those with diarrhea tested positive for the virus.  Thus, the association between BCoV and bovine diarrhea is not considered very tight.  While the group has established that there were six strains circulating in the country at the time, it is not clear that this can be considered a major advance.  The organization and use of the English language is somewhat improved in the revised version.  However, this does not enhance the significance of the findings.

Comments on the Quality of English Language

English usage is improved in the revised version.